# LOCAL CLUSTERING GRAPH NEURAL NETWORKS

## ABSTRACT

Graph Neural Networks (GNNs), which benefit various real-world problems and applications, have emerged as a powerful technique for learning graph representations. The depth of a GNN model, denoted by $K$, restricts the receptive field of a node to its $K$-hop neighbors and plays a subtle role in the performance of GNNs. Recent works demonstrate how different choices of $K$ produce a trade-off between increasing representation capacity and avoiding over-smoothing. We establish a theoretical connection between GNNs and local clustering, showing that short random-walks in GNNs have a high probability to be stuck at a local cluster. Based on the theoretical analysis, we propose **L**ocal **C**lustering **G**raph **N**eural **N**etworks (LCGNN), a GNN learning paradigm that utilizes local clustering to efficiently search for small but compact subgraphs for GNN training and inference. Compared to full-batch GNNs, sampling-based GNNs and graph partition-based GNNs, LCGNN performs comparably or even better, achieving state-of-the-art results on four Open Graph Benchmark (OGB) datasets. The locality of LCGNN allows it to scale to graphs with 100M nodes and 1B edges on a single GPU.

## 1 INTRODUCTION

Recent emergence of the *Graph Neural Networks (GNNs)*, exemplified by models like ChebyNet (Defferrard et al., 2016), GCN (Kipf & Welling, 2017), GraphSAGE (Hamilton et al., 2017), GAT (Veličković et al., 2018), and GIN (Xu et al., 2019), has drastically reshaped the landscape of the graph learning research. These methods generalize traditional deep learning algorithms to model graph-structured data by combining graph propagation and neural networks. Despite its conceptual simplicity, GNNs have reestablished the new state-of-the-art methods in various graph learning tasks, such as node classification, link prediction, and graph classification (Hu et al., 2020; Dwivedi et al., 2020), also served as key contributors to many real-world applications, such as recommendation system (Ying et al., 2018), smart transportation (Luo et al., 2020), visual question answering (Teney et al., 2017) and molecular de-novo design (You et al., 2018).

With the growth of real-world social and information networks (Leskovec et al., 2005), there is an urgent need to scale GNNs to massive graphs. For example, the recommendation systems in Alibaba (Zhu et al., 2019) and Pinterest (Ying et al., 2018) require training and inferring GNNs on graphs with billions of edges. Building such large-scale GNNs, however, is a notoriously expensive process. For instance, the GNN models in Pinterest are trained on a 500GB machine with 16 Tesla K80 GPUs, and served on a Hadoop cluster with 378 d2.8xlarge Amazon AWS machines.

Although one may think model parameters are the main contributors to the huge resource consumption of GNNs, previous work (Ma et al., 2019) suggests the main bottleneck actually comes from the entanglement between graph propagation and neural networks, which leads to a large and irregular computation graph for GNNs. This problem is further exacerbated by the small-world phenomenon (Watts & Strogatz, 1998), i.e., even a relatively small number of graph propagation can involve full-graph computation. For example, in Facebook college graphs of John Hopkins (Traud et al., 2012), the 2-hop neighbors of node 1, as shown in Fig. 1a, covers 74.5% of the whole graph.

A common strategy to reduce the overhead of GNNs is to make the graph smaller but may bring side effects. For instance, graph sampling techniques, such as neighborhood sampling in Graph-SAGE (Hamilton et al., 2017), may lead to the high variance issue (Chen et al., 2018a). Alternatively, graph partition techniques, such as METIS (Karypis & Kumar, 1998) that adopted by Cluster-GCN (Chiang et al., 2019) and AliGraph (Zhu et al., 2019), essentially involves extra full-

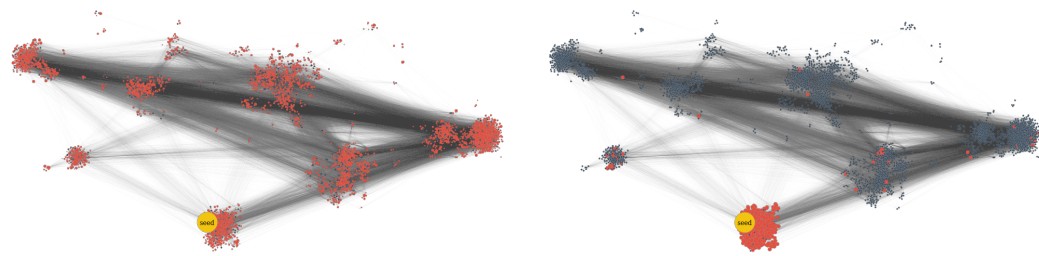

(a) 2-hop neighbors of node 1 covers 74.5% of the graph.      (b) A local cluster around node 1.

Figure 1: Motivating examples from the John Hopkins graph.

graph computation for preprocessing. Besides reducing graph size, a recent attempt (Wu et al., 2019) to scale up GNNs is to decouple graph propagation and neural networks. However, such simplification may sacrifice certain performances.

In this work, we explore a theoretically guaranteed strategy, *local clustering* (Spielman & Teng, 2013; Andersen et al., 2006), intending to design a lightweight, effective and scalable GNN framework. We establish a connection between GNNs and local clustering theory, showing that the graph propagation in GNNs (i.e., short random-walk) has a high probability to be stuck at a local cluster (a.k.a, community), and the escaping probability is proportional to the conductance of the local cluster. We propose Local Clustering Graph Neural Networks (LCGNN), which utilizes local clustering algorithms to seek local and dense subgraphs (e.g., Fig. 1b) for GNN training and inference. Different from full-batch and graph partition-based methods, LCGNN does not incur full-graph processing and can be conducted locally. Compared to various baselines, LCGNN achieves state-of-the-art results in four Open Graph Benchmark (Hu et al., 2020) datasets. Moreover, the locality nature of LCGNN allows it to scale to graphs with 100M nodes and 1B edges on a single GPU.

The rest of the paper is organized as follows. Section 2 gives a brief background summary followed by a survey of related works in section 3. In section 4 and section 5, we establish the connection between GNNs and local clustering, and then describe our LCGNN framework. Section 6 presents the experimental results and ablation study. Finally, we concludes this work in section 7.

## 2 BACKGROUND

In this section, we bring the necessary background about graph, graph convolutional networks (GCN), (lazy) random walk on graphs, and graph conductance.

**Graph Notations** The graph $G = (V, E, \boldsymbol{A})$ consists of $|V| = n$ nodes and $|E| = m$ edges. $\boldsymbol{A} \in \mathbb{R}_+^{n \times n}$ is the adjacency matrix where its entry $\boldsymbol{A}(i, j)$, if nonzero, denote there is an edge between node $i$ and $j$ with edge weight $\boldsymbol{A}_{ij}$. In this work, we assume the input graph is undirected and unweighted, and our analysis can be generalized to the weighted graph case easily. For undirected graph, the degree matrix $\boldsymbol{D} \triangleq \mathrm{diag}(d(1), \cdots, d(n))$ is a diagonal matrix where $d(i) \triangleq \sum_j \boldsymbol{A}(i, j)$ is the degree of node $i$. Moreover, each node in $G$ is associated with a $F$-dimensional feature vector, denoted by $\boldsymbol{x}_i \in \mathbb{R}^F$. The entire feature matrix $\boldsymbol{X} \in \mathbb{R}^{n \times F}$ is the concatenation of node feature vectors. There are two matrices that play importance roles in the design and analysis of GCN (Kipf & Welling, 2017) — the normalized graph Laplacian $\boldsymbol{L} \triangleq \boldsymbol{D}^{-1/2} \boldsymbol{A} \boldsymbol{D}^{-1/2}$ and the random walk transition probability matrix $\boldsymbol{P} \triangleq \boldsymbol{A} \boldsymbol{D}^{-1}$. Note that the entry $\boldsymbol{P}(i, j)$ indicates the probability that the random walk goes from node $j$ to node $i$.

**Graph Convolutional Networks (GCN)** GCN (Kipf & Welling, 2017) initializes the node representation as the input feature matrix $\boldsymbol{H}^{(0)} \leftarrow \boldsymbol{X}$, and iteratively apply non-linear transformation and graph propagation on node representation: $\boldsymbol{H}^{(k)} \leftarrow \mathrm{ReLU}\left(\boldsymbol{L}\boldsymbol{H}^{(k-1)}\boldsymbol{W}^{(k)}\right)$, where left multiplying $\boldsymbol{H}^{(k-1)}$ by normalized graph Laplacian $\boldsymbol{L}$ acts as the graph propagation, and right multiplying $\boldsymbol{H}^{(k-1)}$ by $\boldsymbol{W}$ as well as the ReLU (Glorot et al., 2011) activation acts as the non-linear transformation. For the node classification task, a $K$-layer GCN predicts node labels $\boldsymbol{Y}$ with a softmax

classifier: $\boldsymbol{Y} \leftarrow \operatorname{softmax}\left(\boldsymbol{L}\boldsymbol{H}^{(K-1)}\boldsymbol{W}^{(K)}\right)$. Take a two-layer GCN ($K = 2$) as a running example, the predicted node labels $\boldsymbol{Y}$ is defined as $\boldsymbol{Y} \leftarrow \operatorname{softmax}\left(\boldsymbol{L}\mathrm{ReLU}\left(\boldsymbol{L}\boldsymbol{H}^{(0)}\boldsymbol{W}^{(1)}\right)\boldsymbol{W}^{(2)}\right)$.

**Lazy Random Walk** In practice, many GNNs add (weighted) self-loops to the graphs ($\boldsymbol{A} \leftarrow \boldsymbol{A} + \alpha\boldsymbol{I}$, Kipf & Welling (2017); Xu et al. (2019)) or create residual connections (He et al., 2016) in neural networks (Li et al., 2019; Dehmamy et al., 2019). Such techniques can be viewed as variants of lazy random walk on graphs — at every step, with probability $1/2$ the walker stays at the current node (through a self-loop) and with probability $1/2$ the walker travels to a neighbor. The transition matrix of a lazy random walk is $\boldsymbol{M} \triangleq (\boldsymbol{I} + \boldsymbol{A}\boldsymbol{D}^{-1})/2$. In this work, we mainly consider lazy random walk. Because it has several desired properties and it is consistent with the actual situation.

**Graph Conductance** For an undirected unweighted graph $G = (V, E, \boldsymbol{A})$, the graph volume of any non-empty node set $S \subset V$ is defined as $\mathrm{vol}(S) \triangleq \sum_{i \in S} d(i)$, which measures the total number of edges incident from $S$. The *conductance* of a non-empty node set $S \subset V$ is defined as $\Phi(S) \triangleq \frac{\sum_{i \in S}\sum_{j \in V - S}\boldsymbol{A}(i,j)}{\min\left(\mathrm{vol}(S), \mathrm{vol}(V-S)\right)}$. Roughly speaking, conductance $\Phi(S)$ is the ratio of the number of edges across $S$ and $V - S$ to the number of edges incident from $S$, measuring the clusterability of a subset $S$. Low conductance indicates a good cluster because its internal connections are significantly richer than its external connections. Although it is NP-hard to minimize conductance (Šíma & Schaeffer, 2006), there have been theoretically-guaranteed approximation algorithms that identify clusters near a given node that satisfy a target conductance condition, such as Spielman & Teng (2013); Andersen et al. (2006); Chung (2007).

## 3 RELATED WORK

The design of scalable GNNs has attracted wide attention from the machine learning community. We review related work from three perspectives: (1) full-batch GNNs with co-design of systems and algorithms; (2) sampling-based GNNs; (3) graph partition-based GNNs.

**Full-batch GNNs** A full-batch GNN takes a whole graph as input for forward and backward. Consequently, its computational cost is proportional to the graph size. Earlier GNN models (Kipf & Welling, 2017; Veličković et al., 2018) evaluated on relatively small graphs, thus can be trained in a full-batch manner. Scaling full-batch GNNs to large graphs requires the co-design of ML systems and ML algorithms (Jiang et al., 2020; Zhang et al., 2020; Ma et al., 2019). For example, NeuGraph (Ma et al., 2019) runs full-batch GNN models on a graph with 8.6M nodes and 231.6M edges on an eight-P100-GPU server. SGC (Wu et al., 2019) is another attempt at full-batch GNN. It simplifies GCN by conducting graph propagation and classification separately and efficiently. However, such simplification may sacrifice performance in some downstream tasks.

**GNNs based on Graph Sampling** GraphSAGE (Hamilton et al., 2017) first proposed the idea of neighborhood sampling, and later it was applied in a real-world recommendation system by Pin-SAGE (Ying et al., 2018). At each GNN layer, GraphSAGE computes node representation by first down-sampling its neighborhoods and then aggregating the sampled ones. As a randomized algorithm, Neighborhood Sampling was further improved by FastGCN (Chen et al., 2018b), Stochastic GCN (Chen et al., 2018a) and Adaptive Sampling (Huang et al., 2018) for variance reduction. A recent work about sampling-based GNN is GraphSAINT (Zeng et al., 2020), which samples subgraphs (Leskovec & Faloutsos, 2006) and run full-batch GNN on sampled subgraphs.

**GNNs based on Graph Partition** Cluster-GCN (Chiang et al., 2019) is the most related work to ours. Cluster-GCN adopts global graph partition algorithms, METIS (Karypis & Kumar, 1998), to partition the input graph into subgraphs, and run a GNN on each subgraph. A similar idea was also proposed in AliGraph (Zhu et al., 2019). However, global graph partition algorithms involve additional whole graph computation. Moreover, global graph partition algorithms are vulnerable to dynamic and evolving graphs (Xu et al., 2014; Vaquero et al., 2014), with nodes and edges being constantly added and removed, which are very common in real-world applications.

# 4 SHORT RANDOM WALK AS LOCAL CLUSTERING

Most GNNs adopt short random walks to explore a graph. For example, the default 2-layer GCN in Kipf & Welling (2017) can be viewed as enumerating all length-2 paths and aggregating them with a neural network; Hamilton et al. (2017) uses a 2-hop neighborhood sampling method, a variant of 2-hop random walk, to sample neighbors in each GraphSAGE layer. GraphSAINT (Zeng et al., 2020) samples subgraphs by 2-hop random walks and then build a full-batch GCN on them. SGC (Wu et al., 2019) conducts 2-hop feature propagation and then apply node-wise logistic regression.

We reveal the theoretical connection between short random walk and local clustering. To be more formal, let $q^{(K)}$ be the $K$-th step lazy random-walk distribution starting from an arbitrary node $u$ according to transition probability matrix $M$, i.e., $q^{(K)} \leftarrow M^K \mathbf{1}_u$. We want to study the probability vector $q^{(K)}$ in terms of $K$, especially when $K$ is small (e.g., $K = 2$). Due to the the small world phenomenon (Watts & Strogatz, 1998), for most social/information networks, $q^{(K)}$ can have $O(n)$ non-zeros, even $K$ is small, e.g., $K = 2$ or 3. However, the following theorem shows that the probability that a random walk escaping from a local cluster can be bounded by its conductance:

**Theorem 1** (Escaping Mass, Proposition 2.5 in Spielman & Teng (2013)). *For all $K \geq 0$ and all $S \subset V$, the probability that any $K$-step lazy random walk staring in $S$ escapes $S$ is at most $K\Phi(S)/2$. I.e., the escaping probability satisfies $q^{(K)}(V - S) \leq K\Phi(S)/2$.*

The key point of Theomre 1 is to relate the $K$-th step random-walk probability to graph conductance — for a node $u$, suppose there exists a subset $S$ such that (1) $u \in S$ and (2) $\Phi(S)$ is small (low conductance), Theorem 1 guarantees that the probability that a lazy random walk starting from node $u$ is very likely to be stuck at $S$, revealing the following facts and potential problems of existing GNNs: (1) for full-batch GNNs, although its receptive field induced by $K$-hop neighbors may cover the whole graph, most probability mass still concentrates around a local cluster (if exists), and the remaining probabilities (i.e., escaping mass) are small and bounded. Consequently, the computation cost of full-batch GNNs can be largely reduced; (2) Sampling-based methods can be viewed as a randomized and implicit version of finding a local clustering, however, with their sample-efficiency and variance non-guaranteed. The above facts encourage us to design local clustering-based GNNs.

A crucial question about the above analysis is the existence of a low-conductance $S$ for every node $u$ (or most nodes in the graph). This is generally not true for arbitrary graphs, e.g., a complete graph. However, evidence from network science and social science agrees with our assumption. For example, (1) Many networks of interest in the sciences are found to divide naturally into communities (Girvan & Newman, 2002; Newman, 2006); (2) Real-world social networks consist of compact communities with size scale of around 100 nodes (Leskovec et al., 2009); (3) Roughly 150 individuals are the upper limit on the size of a well-functioning human community (Dunbar, 1998).

# 5 LOCAL CLUSTERING GRAPH NEURAL NETWORKS (LCGNN)

The analysis in section 4 lays the theoretical foundation of the design of our LCGNN framework. In the section, we formally introduce LCGNN. Roughly speaking, our framework consists of two steps. In the first step, for each node $u \in V$, we run local clustering to produces a local cluster $S_u$ surrounding it. In the second step, we feed the subgraph induced by $S_u$ to a GNN encoder.

## 5.1 LOCAL CLUSTERING

Local clustering algorithms find a small cluster near given seed(s). Different from global graph partition methods involving full-graph computation, local clustering conducts local exploration in the graph and its running time depends only on the size of the output cluster. Over the past two decades, many local clustering algorithms have been developed (Spielman & Teng, 2013; Andersen et al., 2006; Chung, 2007; Li et al., 2015; Kloster & Gleich, 2014; Kloumann & Kleinberg, 2014; Whang et al., 2013; Yin et al., 2017; Fountoulakis et al., 2019). In this works, we mainly focus on PPR-Nibble (Andersen et al., 2006), one of the most popular spectral-based local clustering algorithms among the above methods. As its name indicates, PPR-Nibble adopts the personalized PageRank (PPR) vector for local clustering. The PPR vector $p_u$ of a node $u$ is given by equation $p_u = \alpha \mathbf{1}_u + (1-\alpha) P p_u$, which is the stationary distribution of the following random walk: at each

---

**Algorithm 1:** Approximate-PPR.

---

1 **Input** Graph $G = (V, E, \boldsymbol{A})$, seed node $u$, teleportation parameter $\alpha$, tolerance $\epsilon$;
2 **Output** An $\epsilon$-approximate PPR vector $\widetilde{\boldsymbol{p}}_u$;
3 $\widetilde{\boldsymbol{p}}_u \leftarrow \boldsymbol{0}; \boldsymbol{r} \leftarrow \boldsymbol{1}_u$;
4 **while** $\boldsymbol{r}(v)/d(v) \geq \epsilon$ *for some* $v \in V$ **do**
5     $\rho \leftarrow \boldsymbol{r}(v) - \frac{\epsilon}{2}d(v); \widetilde{\boldsymbol{p}}_u(v) \leftarrow \widetilde{\boldsymbol{p}}_u(v) + \alpha\rho; \boldsymbol{r}(v) \leftarrow \frac{\epsilon}{2}d(v)$;
6     **for** *each* $(v, u) \in E$ **do**
7        $\left\lfloor \boldsymbol{r}(u) \leftarrow \boldsymbol{r}(u) + \frac{\boldsymbol{A}(v,u)}{d(v)}(1 - \alpha)\rho \right.$;

8 **return** $\widetilde{\boldsymbol{p}}_u$;

---

**Algorithm 2:** PPR-Nibble.

---

1 **Input** Graph $G = (V, E, \boldsymbol{A})$, seed node $u$, teleportation parameter $\alpha$, tolerance $\epsilon$;
2 **Output** A local cluster $S \subset V$;
3 $\widetilde{\boldsymbol{p}}_u \leftarrow$ Approximate-PPR$(G, u, \alpha, \epsilon)$;
4 $\sigma_i \leftarrow i-$th largest entry of $\boldsymbol{D}^{-1}\widetilde{\boldsymbol{p}}_u$;
5 **return** $S \leftarrow \arg\min_{S_\ell} \Phi(S_\ell)$, *where* $S_\ell = \{\sigma_1, \cdots, \sigma_\ell\}$;

---

step of the random walk, with probability $\alpha$ the walker teleports back to the node $u$, and with probability $1 - \alpha$ the walker performs a normal random walk. However, PPR vector $\boldsymbol{p}_u$ is a dense vector and thus computationally expensive. Andersen et al. (2006) developed an efficient algorithm, named Approximate-PPR to compute its sparse approximation $\widetilde{\boldsymbol{p}}_u$ so that $|\boldsymbol{p}_u(v)/d(v) - \widetilde{\boldsymbol{p}}_u(v)/d(v)| \leq \epsilon$ for each node $v$. As shown in Algorithm 1, the key idea is to gradually push probabilities from a residual vector $\boldsymbol{r}$ to approximate PPR vector $\widetilde{\boldsymbol{p}}_u$ (Line 5-7 of Algorithm 1). After computing the approximate PPR vector $\widetilde{\boldsymbol{p}}_u$, a *sweep* procedure is then adopted to extract a cluster $S$ with small conductance $\Phi(S)$. More formally, the sweep procedure first sort nodes according to $\boldsymbol{D}^{-1}\widetilde{\boldsymbol{p}}_u$ in descending order (Line 4 of Algorithm 2), and then evaluate the conductance of each node prefix in the sorted list and output the one with smallest conductance (Line 5 of Algorithm 2).

Note that PPR-Nibble is a local algorithm (Spielman & Teng, 2013) with theoretical guarantee — (1) The input to the algorithm is a starting node $u$; (2) At each step of Approximate-PPR in Algorithm 1, it only examines nodes connected to those it has seen before. The following theorems characterize the complexity and error bounds of Approximate-PPR of PPR-Nibble, respectively.

**Theorem 2** (Lemma 2 in Andersen et al. (2006)). *Algorithm 1 runs in time* $O\left(\frac{1}{\alpha\epsilon}\right)$. *and the number of non-zeros in* $\widetilde{\boldsymbol{p}}_u$ *satisfies* $nnz(\widetilde{\boldsymbol{p}}_u) \leq \frac{1}{\alpha\epsilon}$.

**Theorem 3** (Theorem 1 in Zhu et al. (2013); Theorem 4.3 in Yin et al. (2017)). *Let* $S \subset V$ *be some unknown targeted cluster, we are trying to retrieve from an unweighted graph. Let* $\eta$ *be the inverse mixing time of the random walk on the subgraph induced by* $S$. *Then there exists* $S^g \subseteq S$ *with* $vol(S^g) \geq vol(S)/2$, *such that for any seed* $u \in S^g$, *Algorithm 2 with* $\alpha = \Theta(\eta)$ *and* $\epsilon \in \left[\frac{1}{10\,vol(T)}, \frac{1}{5\,vol(T)}\right]$ *outputs a set* $S$ *with* $\Phi(S) \leq \widetilde{O}\left(\min\left\{\sqrt{\Phi(T)}, \Phi(T)/\sqrt{\eta}\right\}\right)$.

### 5.2 LOCAL CLUSTER ENCODER

For each node $u \in V$, PPR-Nibble in Algorithm 2 produces a local cluster $S_u \subset V$ with $|S_u| \leq \frac{1}{\alpha\epsilon}$. We denote $G_u$ to be the subgraph induced by the cluster $S_u$, which is then encoded to a hidden representation via an encoder (usually a GNN model): $\boldsymbol{h}_u \leftarrow \text{ENCODER}(G_u)$. The encoded hidden representation can be further used for various graph learning tasks. For the node classification task, we predict the label of node $u$ with a softmax classifier: $y_u \leftarrow \text{softmax}(\boldsymbol{W}\boldsymbol{h}_u + \boldsymbol{b})$; For the link prediction task, we measure the likelihood of a link $e = (u, v)$ by first element-wisely multiplying $\boldsymbol{h}_u$ and $\boldsymbol{h}_v$ and then feeding it to a MLP encoder, i.e., $y_e \leftarrow \text{MLP}(\boldsymbol{h}_u \odot \boldsymbol{h}_v)$.

The choice of the encoder is flexible. In this work, we mainly examine four candidate encoders:

**GCN/GAT/GraphSAGE encoders** Our first candidate encoders are traditional GNNs such as GCN, GAT, and GraphSAGE. We denote them as LCGNN-GCN/-GAT/-SAGE, respectively.

Table 1: Statistics of datasets for node classification and link prediction tasks.

| Category | Name | #Nodes | #Edges | Split Ratio | Metric |
|---|---|---|---|---|---|
| **Node** ogbn- | products | 2,449,029 | 61,859,140 | 10/02/88 | Accuracy |
| | arxiv | 169,343 | 1,166,243 | 54/18/28 | Accuracy |
| | papers100M | 111,059,956 | 1,615,685,872 | 78/8/14 | Accuracy |
| **Link** ogbl- | ppa | 576,289 | 30,326,273 | 70/20/10 | Hits@100 |
| | collab | 235,868 | 1,285,465 | 92/4/4 | Hits@50 |
| | citation | 2,927,963 | 30,561,187 | 98/1/1 | MRR |

**Transformer Encoder**  We also examine a more complex and powerful encoder based on Transformer (Vaswani et al., 2017). Our hypothesis is that low conductance subgraphs extracted by local clustering have such rich internal connections that we can almost treat them as complete graphs. Thus we adopt the Transformer encoder whose attention mechanism allows dense interaction within a subgraph. We initialize the positional embedding in Transformer as the pre-trained Node2vec embedding on the input graph. We denote the Transformer-based encoder as LCGNN-Transformer.

## 6 EXPERIMENTS

In this section, we conduct experiments on two major tasks of graph learning, node classification and link prediction. For each task, we use the datasets from Open Graph Benchmark (OGB) (Hu et al., 2020), which presents significant challenges of scalability to large-scale graphs and out-of-distribution generalization. The dataset statistics are summarized in Table 1, Another graph task, graph classification, is not explored in our experiments because it is unnecessary to utilize local clustering for small graphs with only hundreds of nodes. The average and standard deviation of test performance under 10 different seeds are reported in all experiments. For the local clustering algorithm, we use the software provided by Fountoulakis et al. (2018). We set $\alpha = 0.15$ in Approximate-PPR, and constraint the maximum cluster size to be 64 or 128 in the PPR-Nibble step, i.e., the sweep procedure only examines the prefix of first 64 (128) nodes in Algorithm 2. Detailed hyper-parameter configuration of LCGNN can be found in Appendix A.2.

### 6.1 NODE CLASSIFICATION

Node classification datasets include *products*, *arxiv*, and *papers100M* at different scales. We train LCGNN on a single GPU on all three datasets. Limited by space, the results of the arxiv dataset are reported in the Appendix A.1 because relatively small datasets are not our target scenario.

**Baselines.**  The OGB team provides MLP, Node2vec (Grover & Leskovec, 2016), GCN (Kipf & Welling, 2017), GraphSAGE (Hamilton et al., 2017) as the common baselines for products and arxiv datasets. For the large-scale papers100M dataset, the OGB team only provides MLP, Node2vec, and SGC (Wu et al., 2019). Other teams and researchers also contribute numerous models to the leaderboards: For the products dataset, three GAT-based models with different mini-batch training techniques are also reported. DeeperGCN (Li et al., 2020) explores how to design and train deep GCNs. UniMP (Shi et al., 2020) is a most recent model[1] which combines feature propagation and label propagation.

**Results.**  The results of products and papers100M datasets are listed in the Table 2 and Table 3, respectively. In papers100M dataset, SGC (Wu et al., 2019) is the only reported GNN model that can handle this large-scale dataset with more than 1 billion edges. SGC gets better performance than Node2vec and MLP due to the expressive power of (simplified) graph convolution. Compared with SGC, LCGNN uses a semi-supervised manner and can learn feature transformation in the training procedure. Our proposed LCGNN obtains better performance than SGC with 2.73% absolute improvement, which shows stronger expressiveness of our model. In products dataset, our LCGNN (rank 2 in Table 2) gets comparable results with other state-of-the-art GNN models. The arxiv dataset is relatively small and well-tuned full-bath GNNs achieve the best results. Our

---

[1]UniMP was submitted to OGB leaderboard on Sep 8, 2020, in one month before ICLR 2021 deadline.

Table 2: ogbn-products leaderboard (collected on Oct. 1, 2020). Limited by paper space, we only list top results. * denotes that the results are submitted in one month before ICLR 2021 deadline.

| Method | Test Accuracy | Validation Accuracy | #Params |
|---|---|---|---|
| GCN | $0.7564 \pm 0.0021$ | $0.9200 \pm 0.0003$ | 103,727 |
| GraphSAGE | $0.7850 \pm 0.0014$ | $0.9224 \pm 0.0007$ | 206,895 |
| ClusterGCN (SAGE aggr) | $0.7897 \pm 0.0033$ | $0.9212 \pm 0.0009$ | 206,895 |
| GraphSAINT (SAGE aggr) | $0.7908 \pm 0.0024$ | $0.9162 \pm 0.0008$ | 206,895 |
| NeighborSampling (GAT aggr) | $0.7945 \pm 0.0059$ | - | 1,751,574 |
| GraphSAINT (GAT aggr) | $0.8027 \pm 0.0026$ | - | 331,661 |
| DeeperGCN | $0.8098 \pm 0.0020$ | $0.9238 \pm 0.0009$ | 253,743 |
| UniMP* | $\mathbf{0.8256 \pm 0.0031}$ | $\mathbf{0.9308 \pm 0.0017}$ | 1,475,605 |
| LCGNN-GCN | $0.7683 \pm 0.0035$ | $0.9259 \pm 0.0009$ | 132,015 |
| LCGNN-SAGE | $0.7858 \pm 0.0027$ | $0.9254 \pm 0.0009$ | 194,479 |
| LCGNN-GAT | $0.8080 \pm 0.0025$ | $0.9285 \pm 0.0013$ | 329,263 |
| LCGNN-Transformer | $0.8131 \pm 0.0059$ | $0.9249 \pm 0.0008$ | 85,167 |

Table 3: ogbn-papers100M leaderboard (collected on Oct. 1, 2020)

| Method | Test Accuracy | Validation Accuracy | #Params |
|---|---|---|---|
| MLP | $0.4724 \pm 0.0031$ | $0.4960 \pm 0.0029$ | 144,044 |
| Node2vec | $0.5560 \pm 0.0023$ | $0.5807 \pm 0.0028$ | 14,215,818,412 |
| SGC | $0.6329 \pm 0.0019$ | $0.6648 \pm 0.0020$ | 144,044 |
| LCGNN-Transformer | $\mathbf{0.6602 \pm 0.0007}$ | $\mathbf{0.6930 \pm 0.0006}$ | 437,036 |

LCGNN gets comparable results to full-batch GNNs and achieves better results than sampling-based GNNs (such as GAT with neighbor sampling), as shown in the Table 7 in the Appendix A.1.

**Ablation Study.** Table 2 suggests that LCGNN-GCN and LCGNN-SAGE surpass the corresponding full-batch GCN and GraphSAGE. Furthermore, LCGNN-SAGE and LCGNN-GAT perform competitively or even better on products dataset comparing to corresponding GraphSAGE and GAT models with other training and sampling techniques, including Neighborhood Sampling (Hamilton et al., 2017), ClusterGCN (Chiang et al., 2019), and GraphSAINT (Zeng et al., 2020).

## 6.2 LINK PREDICTION

We evaluate LCGNN on three link prediction tasks — *ppa*, *collab*, and *citation*. We use a single GPU to train on the collab dataset and use multi-GPUs to train on the ppa and citation datasets (5 GPUs for ppa and 4 GPUs for citation).

**Baselines.** The OGB team provides Matrix Factorization, Node2vec (Grover & Leskovec, 2016), GCN (Kipf & Welling, 2017), GraphSAGE (Hamilton et al., 2017) as the common baselines. For the citation dataset, GCN/SAGE-based models with three different mini-batch training techniques are also provided by the OGB team. Other researchers also contribute some state-of-the-art models to the leaderboards. DeepWalk (Perozzi et al., 2014) is submitted by other researchers using DGL (Wang et al., 2019). LRGA+GCN (Puny et al., 2020) is a recently proposed model which aligns 2-folklore Weisfeiler-Lehman algorithm to improve the generalization of GNNs.

Table 4: ogbl-ppa leaderboard (collected on Oct. 1, 2020)

| Method | Test Hits@100 | Validation Hits@100 | #Params |
|---|---|---|---|
| GraphSAGE | $0.1655 \pm 0.0240$ | $0.1724 \pm 0.0264$ | 424,449 |
| GCN | $0.1867 \pm 0.0132$ | $0.1845 \pm 0.0140$ | 278,529 |
| Node2vec | $0.2226 \pm 0.0083$ | $0.2253 \pm 0.0088$ | 73,878,913 |
| DeepWalk | $0.2888 \pm 0.0153$ | - | 150,138,741 |
| Matrix Factorization | $0.3229 \pm 0.0094$ | $0.3228 \pm 0.0428$ | 147,662,849 |
| LRGA + GCN | $0.3426 \pm 0.0160$ | - | - |
| LCGNN-Transformer | $\mathbf{0.3535 \pm 0.0115}$ | $\mathbf{0.3569 \pm 0.0129}$ | 306,304 |

Table 5: ogbl-collab leaderboard (collected on Oct. 1, 2020)

| Method | Test Hits@50 | Validation Hits@50 | #Params |
|---|---|---|---|
| Matrix Factorization | 0.3886±0.0029 | 0.4896±0.0029 | 60,514,049 |
| GCN | 0.4475±0.0107 | 0.5263±0.0115 | 296,449 |
| GraphSAGE | 0.4810±0.0081 | 0.5688±0.0077 | 460,289 |
| Node2vec | 0.4888±0.0054 | 0.5703±0.0052 | 30,322,945 |
| DeepWalk | 0.5037±0.0034 | - | 61,390,187 |
| LRGA + GCN | 0.5221±0.0072 | 0.6088±0.0059 | 1,069,489 |
| LCGNN-GCN | 0.5351 ± 0.0202 | 0.6009 ± 0.0190 | 50,304 |
| LCGNN-SAGE | 0.5315 ± 0.0124 | 0.5997 ± 0.0141 | 99,840 |
| LCGNN-Transformer | **0.5485 ± 0.0068** | **0.6432 ± 0.0074** | 414,848 |

Table 6: ogbl-citation leaderboard (collected on Oct. 1, 2020)

| Method | Test Hits@50 | Validation Hits@50 | #Params |
|---|---|---|---|
| Matrix Factorization | 0.5316 ± 0.0565 | 0.5311 ± 0.0565 | 281,113,505 |
| Node2vec | 0.5964 ± 0.0011 | 0.5944 ± 0.0011 | 374,911,105 |
| GraphSAINT (GCN aggr) | 0.7943 ± 0.0043 | 0.7933 ± 0.0046 | 296,449 |
| ClusterGCN (GCN aggr) | 0.8021 ± 0.0029 | 0.7999 ± 0.0027 | 296,449 |
| NeighborSampling (SAGE aggr) | 0.8048 ± 0.0015 | 0.8048 ± 0.0015 | 460,289 |
| GraphSAGE | 0.8228 ± 0.0084 | 0.8217 ± 0.0086 | 460,289 |
| DeepWalk | 0.8284 ± 0.0005 | - | 757,943,019 |
| GCN | 0.8456 ± 0.0110 | 0.8449 ± 0.0108 | 296,449 |
| LCGNN-Transformer | **0.8524 ± 0.0046** | **0.8510 ± 0.0046** | 315,264 |

**Results.** The results of ppa, collab, and citation datasets are listed in the Table 4, 5, and 6, respectively. We compare LCGNN with a recently developed model, LRGA+GCN (Puny et al., 2020), as well as traditional baselines. For all three datasets for link prediction, our proposed LCGNN achieves the best results over state-of-the-art models with $0.68\% \sim 2.64\%$ absolute improvements, showing the ability of local clustering and Transfomer encoder to boost link-prediction performance.

**Ablation Study.** We report the results on the collab when the Transformer encoder is replaced with GCN and GraphSAGE encoders. Compared with full-batch GCN and GraphSAGE, our LCGNN-GCN and LCGNN-SAGE obtains much better performance, which suggests the significance of graph local clustering. LCGNN-Transformer gets better results than LCGNN-GCN and LCGNN-SAGE due to the powerful expressiveness of the Transformer encoder.

Overall, not only LCGNN achieves four first places (ogbn-paper100m, ogbl-ppa, ogbl-collab, and ogbl-citation) and one second place (ogbn-products) on OGB datasets, it also improves the scalability of GNN models for large-scale graphs.

# 7 CONCLUSION

In this work, we present Local Clustering Graph Neural Networks (LCGNN), a lightweight, effective, and scalable GNN framework with theoretical guarantees. LCGNN combines local clustering algorithms and graph neural network models to achieve state-of-the-art performance on four Open Graph Benchmark (OGB) datasets. By incorporating local clustering algorithms, LCGNN can run on compact and small subgraphs without conducting full-graph computation, scaling to graphs with 100 million nodes and 1 billion edges on a single GPU. In the future, it would be interesting to try more advanced local clustering algorithms other than the PPR-Nibble. Applying LCGNN in real-world applications, such as the recommendation system, is also a promising direction.

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

## A  APPENDIX

### A.1  EXPERIMENTAL RESULTS

We report the results of the ogbn-arxiv dataset in the Table 7. There are some models that are only evaluated on the smallest dataset (i.e., ogbn-arxiv), including GraphZoom (Deng et al., 2020), GaAN (Zhang et al., 2018), DAGNN (Liu et al., 2020), JKNet (Xu et al., 2018), GCNII (Chen et al., 2020). Most of these models cannot handle ogbn-products with millions of nodes. Our LCGNN models get comparable results with most state-of-the-art GNN models with and without mini-batch training techniques.

Table 7: ogbn-arxiv leaderboard (collected on Oct. 1, 2020). † denotes that the result is run by ourselves. * denotes that the results are submitted in one month before ICLR 2021 deadline.

| Method | Test Accuracy | Validation Accuracy | #Params |
|---|---|---|---|
| MLP | $0.5550 \pm 0.0023$ | $0.5765 \pm 0.0012$ | 110,120 |
| Node2vec | $0.7007 \pm 0.0013$ | $0.7129 \pm 0.0013$ | 21,818,792 |
| GraphZoom (Node2vec) | $0.7118 \pm 0.0018$ | $0.7220 \pm 0.0007$ | 8,963,624 |
| GraphSAGE | $0.7149 \pm 0.0027$ | $0.7277 \pm 0.0016$ | 218,664 |
| GCN | $0.7174 \pm 0.0029$ | $0.7300 \pm 0.0017$ | 142,888 |
| DeeperGCN | $0.7192 \pm 0.0016$ | $0.7262 \pm 0.0014$ | 491,176 |
| GaAN | $0.7197 \pm 0.0024$ | - | 1,471,506 |
| DAGNN | $0.7209 \pm 0.0025$ | $0.7290 \pm 0.0011$ | 43,857 |
| NeighborSampling (GAT aggr)† | $0.7211 \pm 0.0030$ | $0.7308 \pm 0.0014$ | 762,256 |
| JKNet (GCN) | $0.7219 \pm 0.0021$ | $0.7335 \pm 0.0007$ | 89,000 |
| GCNII | $0.7274 \pm 0.0016$ | - | 2,148,648 |
| GCN* | $0.7306 \pm 0.0024$ | $0.7442 \pm 0.0012$ | 238,632 |
| UniMP* | $0.7311 \pm 0.0020$ | $0.7450 \pm 0.0005$ | 473,489 |
| GAT* | $0.7365 \pm 0.0011$ | $0.7504 \pm 0.0006$ | 1,628,440 |
| UniMP_large* | $\mathbf{0.7379 \pm 0.0014}$ | $\mathbf{0.7475 \pm 0.0008}$ | 1,162,515 |
| LCGNN-GCN | $0.7198 \pm 0.0018$ | $0.7357 \pm 0.0012$ | 171,560 |
| LCGNN-SAGE | $0.7210 \pm 0.0025$ | $0.7342 \pm 0.0019$ | 204,328 |
| LCGNN-GAT | $0.7213 \pm 0.0011$ | $0.7346 \pm 0.0011$ | 172,200 |
| LCGNN-Transformer | $0.7222 \pm 0.0022$ | $0.7303 \pm 0.0010$ | 136,936 |

### A.2  EXPERIMENTAL SETUP

#### A.2.1  RUNNING ENVIRONMENT

We run our experiments on a single machine with Intel Xeon CPUs (Platinum 8163 @ 2.50GHz), 330GB memory, and 8 NVIDIA Tesla V100 (16GB). The code is written in Python 3.6. We use PyTorch 1.5.1 on CUDA 10.1 to train our models.

#### A.2.2  HYPERPARAMETER CONFIGURATION

For our models, the optimizer used in our experiments is AdamW (Loshchilov & Hutter, 2019) with $\beta_1 = 0.9$, $\beta_2 = 0.999$, and $eps = 10^{-8}$. For LCGNN-GCN/SAGE/GAT, we use this optimizer with no warmup steps. But for LCGNN, we use the following learning rate scheduler with warmup steps, similar to Transformer (Vaswani et al., 2017) except an extra hyper-parameter lr_scale:

$$\text{lr} = \text{lr\_scale} \cdot d_{\text{model}}^{-0.5} \cdot \min(\text{step\_num}^{-0.5}, \text{step\_num} \cdot \text{warmup\_steps}^{-1.5})$$

We use the wandb (Biewald, 2020) tool to help track experiments and search the hyperparameters. The final hyper-parameters used for our models are listed in the Table 8 and Table 9.

Table 8: Hyper-parameters for LCGNN-Transformer on OGB datasets. * denotes that it is trained on multi-GPUs of a single machine (5 GPUs for ppa and 4 GPUs for citation).

| Hyper-parameters | ogbn- | | | ogbl- | | |
|---|---|---|---|---|---|---|
| | products | arxiv | papers100M | ppa | collab | citation |
| number of layers | 3 | 5 | 4 | 3 | 4 | 3 |
| number of heads | 1 | 1 | 2 | 1 | 1 | 1 |
| maximum cluster size | 64 | 64 | 64 | 64 | 128 | 64 |
| hidden size | 64 | 64 | 128 | 128 | 128 | 128 |
| input dropout | 0.2 | 0.2 | 0.2 | 0.1 | 0.1 | 0.1 |
| hidden dropout | 0.5 | 0.5 | 0.4 | 0.4 | 0.4 | 0.4 |
| batch size | 256 | 256 | 512 | 5120* | 128 | 6000* |
| lr scale | 1.0 | 1.0 | 1.0 | 2.0 | 0.5 | 1.0 |
| weight decay | 0.05 | 0.05 | 0.0005 | 0.0 | 0.0005 | 0.05 |
| warmup steps | 20000 | 10000 | 10000 | 5000 | 20000 | 10000 |

Table 9: Hyper-parameters for LCGNN-GCN/SAGE/GAT on OGB datasets

| Hyper-parameters | ogbn-products | | | ogbn-arxiv | | | ogbl-collab | |
|---|---|---|---|---|---|---|---|---|
| | GCN | SAGE | GAT | GCN | SAGE | GAT | GCN | SAGE |
| number of layers | 4 | 4 | 3 | 5 | 4 | 5 | 3 | 3 |
| number of heads | - | - | 1 | - | - | 1 | - | - |
| maximum cluster size | 128 | 128 | 128 | 128 | 128 | 128 | 128 | 128 |
| hidden size | 128 | 128 | 256 | 128 | 128 | 128 | 128 | 128 |
| hidden dropout | 0.3 | 0.3 | 0.4 | 0.4 | 0.4 | 0.3 | 0.0 | 0.0 |
| batch size | 512 | 512 | 512 | 512 | 512 | 512 | 512 | 512 |
| learning rate | 0.002 | 0.002 | 0.002 | 0.004 | 0.002 | 0.002 | 0.001 | 0.001 |
| weight decay | 0.0005 | 0.0005 | 0.005 | 0.0005 | 0.0 | 0.005 | 0.0 | 0.0 |
| batch norm | True | True | True | True | True | True | True | True |
| residual connection | True | True | True | True | True | True | False | False |

