# OpenReview forum: "Local Clustering Graph Neural Networks"
_ICLR.cc/2021/Conference — Reject_

### Official Review · AnonReviewer1 · 2020-10-18
**Interesting idea but the methodology and evaluation are unsatisfying.**

**Rating:** 4
**Confidence:** 5

**Review:**

Summary:
The authors proposed to first extract a subgraph $G_u$ for each node $u$. Then use GNNs to extract the hidden representation $h_u = GNN(G_u)$. The subgraph extraction uses PPR-Nibble with is a conductance based local clustering method.

Pros:
1.	The model seems to be more efficient compare to GNN with global clustering method such as Cluster-GCN.
2.	Using local clustering method to determine the subgraph for each node might be better than random neighborhood sampling in some cases.

Cons:
1.	The intuition on using short random walk is problematic. (See detailed comments)
2.	The experimental results are insufficient to verify the proposed method.
3.	Insufficient related work on local clustering and PageRank-based methods.
4.	While the authors claim that they have theoretical analysis, all stated theorems are from the other papers. I do not aware of any theoretical contribution as claimed by the authors.

Detailed comments:

The main weakness of the paper is the claim that short random walk is sufficient to extract topology information from graph. This contradicts to both [1] and [2]. In [1], the authors used Personalized PageRank (PPR) to build their GNN (APPNP) which shows that large propagation step is helpful ($K=10-20$). Notably, they show that using $K\geq 10$ gives significant improvement on Cora and PubMed dataset for node classification problem compare to using $K\leq 5$. On the other hand, the authors of [2] study the generalized PageRank method for local clustering (seed-set community detection) problem. They show that using large step propagation ($K\geq 10$) leads to better local clustering performance compare to small steps. These works show that using merely short random walk is insufficient to fully extract the topological information from graphs and thus the claim by the authors seems questionable to me.

The other weakness of this paper is their experimental results are insufficient to verify the proposed method outperforms the sampling-base method. For example, in Table 2 the test accuracy of LCGNN-SAGE is actually lower than GraphSAINT (SAGE aggr). Even if we compare LCGNN-GAT with GraphSAINT(GAT-aggr) the gain is not obvious. The model proposed by the authors that has the best performance is LCGNN-Transformer. However, it is not clear whether the performance gain is due to the local clustering procedure or the transformer.

Minor comments:
As the authors mentioned, the local clustering methodology is only reasonable for graphs that has low conductance with respect to all "cluster" (nodes with same label). This is exactly the "Homophily principle" which is true for most of the popular benchmark datasets such as citation networks (Core, Citeseer and PubMed). However, as pointed out in [3], there are also practical graphs that are heterophilic or low homophily. Although this is not the main theme of this paper, the existence of heterophilic graph should not be ignored.

Reference:

[1] “Predict then Propagate: Graph Neural Networks meet Personalized PageRank,” Klicpera et al., ICLR 2018.

[2] “Optimizing Generalized PageRank Methods for Seed-Expansion Community Detection,” Li et al., NeurIPS 2019.

[3] “Geom-GCN: Geometric Graph Convolutional Networks,” Pei et al., ICLR 2020.

---

> ### Author Response · Authors · 2020-11-24
> **Response to AnonReviewer1**
>
> Thank you for your comments. We would also like to clarify a couple of points.
>
> **[Regarding Short Random Walks]** We believe there are some misunderstandings. We NEVER claim that short random walks are sufficient to extract topology information from graphs. The motivation of this work is the FACT that many existing and widely adopted GNNs are based on short random walks like GCN and GraphSAGE, and we want to analyze the behavior of these methods.
>
> **[Regarding [1] and [2]]** We thank the reviewer for providing the references but also want to clarify a couple of points.
>
> As for reference [1],  the behavior of PPR could be very different from that of vanilla random walks. Intuitively, the convergence of PPR is much slower than vanilla random walks and thus PPR requires more propagation steps to explore a graph. In this work, as stated in Section 4, our focus is to analyze the behavior of GNNs based on vanilla random walks.
>
> As for reference [2], the reviewer points out that large step propagation leads to better local clustering performance. However,  this is exactly what we have done in our work. Actually, when using PPR-Nibble for local clustering, we didn’t restrict the number of propagation steps of PPR. As shown in Algorithm 1, PPR-Nibble can propagate many steps until convergence.
>
> **[Regarding Transformer Encoders]** The local clustering part and the Transformer part of our LCGNN framework are just like the two wheels of a cart --- local clustering extracts small but important subgraphs which makes the usage of powerful (but heavy) Transformer encoders affordable and avoids expensive full-graph computation. We also want to emphasize it would be very time/memory expensive to directly use Transformer in other GNN methods, mainly due to the notoriously high cost of Transformers. For example, suppose one wants to run a transformer on ogbn-products graph, then the Transformer encoder has to compute a 2,449,029 by 2,449,029 attention matrix, which is unaffordable.
>
> **[Regarding Theoretical Contribution]** This is NOT a theory paper. We never claim we contribute to the local clustering theory of spectral graph theory. Instead, what we claim is to establish a theoretical connection between GNNs and local clustering, and leverage such connections to design effective and efficient algorithms.
>
> **[Regarding Homophily Principle]** We discuss this assumption at the end of Section 4. Evidence from many network science and social science agrees with our assumption. We agree that there are exceptions as pointed out by [3] and want to thank the reviewer for providing the reference. We will add it in our next version.

---

### Official Review · AnonReviewer3 · 2020-10-27
**A good scalable GNN framework but somewhat incremental**

**Rating:** 5
**Confidence:** 4

**Review:**

[Summary]
In this paper, the authors study the connection between GNNs and local clustering, and find that short random-walks in GNNs have a high probability to be stuck at a local cluster. Based on this, they propose a light and scalable GNN learning framework called LCGNN, which first adopts the local clustering method PPR-Nibble to partition full graph into subgraphs, then use GNN modules on subgraphs for training and inference. The authors evaluate LCGNN on six OGB datasets, and the proposed approach outperforms the competitors on node classification and link prediction tasks.

[Pros]
+ The paper is well organized and easy to follow.
+ The idea of incorporating local clustering algorithms for GNN learning is simple but effective in terms of scalability.

[Cons]
- The proposed method is somewhat incremental since it only adds a local clustering step (with an existing clustering method) before adopting GNN models.
- The experiment is not very sufficient and supportive.
  1. From Table 2, it seems that the improvement of incorporating local clustering (comparing LCGNN-GCN/LCGNN-SAGE with naïve GCN/SAGE) is not very large, and we could not find how much the local clustering improves Transformer. Similar problems appear in Table 3, 4, 6 – most performance improvement seems to come from the adoption of Transformer.
  2. How do clustering hyperparameters $\alpha$ and the maximum cluster size affect the results?
  3. No visualization for the clustering results or the learned node embeddings, which could help understand the effect of local clustering, is presented.
- Time and space complexity of the proposed framework is not provided. Intuitively the proposed method might enjoy better time and space complexity than existing methods, but some theoretical analysis or runtime evaluation would be better to illustrate the main claim of this paper.
- Could LCGNN handle the situation when nodes and edges are constantly added and removed?
- The authors claim that ‘the locality nature of LCGNN allows it to scale to graphs with 100M nodes and 1B edges on a single GPU’ but use 8 NVIDIA Tesla V100 in their experiments. Some results or explanation might be helpful to support this claim.
- There are many typos to be corrected.
  (a) Page 1: covers => cover
  (b) Page 2: we concludes => we conclude
  (c) Page 2: importance roles => important roles
  (d) Page 4: the the small world phenomenon => the small world phenomenon; staring in S => starting in S; I.e., => i.e., (cannot be used at the beginning of the sentence); The key point of Theomre 1 => The key point of Theorem 1; In this works => In this work
Other minor issues: It would be better to highlight the second best value in Table 2.

---

> ### Author Response · Authors · 2020-11-24
> **Response to AnonReviewer3**
>
> Thank you for your comments. We would also like to clarify a couple of points.
>
> **[Regarding Novelty]** Our contributions are two folds. First, we establish a theoretical connection between GNNs and local clustering. Second, we propose a local clustering GNN framework inspired by the theoretical connections, which is able to scale up to very large graphs effectively and efficiently. We believe that a simple but effective method is what we need in practice when applying GNN to real-world large-scale problems.
>
> **[Regarding Transformer Encoders]** The local clustering part and the Transformer part of our LCGNN framework are just like the two wheels of a cart --- local clustering extracts small but important subgraphs which makes the usage of powerful (but heavy) Transformer encoders affordable and avoids expensive full-graph computation. We also want to emphasize it would be very time/memory expensive to directly use Transformer in other GNN methods, mainly due to the notoriously high cost of Transformers. For example, suppose one wants to run a transformer on ogbn-products graph, then the Transformer encoder has to compute a 2,449,029 by 2,449,029 attention matrix, which is unaffordable.
>
> **[Regarding the Effect of alpha and the Maximum Cluster Size]** Thanks for this great question. We fix the $\alpha=0.15$ and search the maximum cluster size from {64, 128}. From Table 8&9 in the Appendix, we can see that for transformer encoders, 64 is better; for gcn/gat/sage encoders, 128 is better. We will add more ablation studies on these hyperparameters.
>
> **[Regarding Visualization]** Thanks for this great suggestion. We show the visualization of a motivating example in Figure 1. A local clustering around node 1 seems much better for representation learning than 2-hop neighbors of node 1.
>
> **[Regarding Complexity]** We preprocess the local clusters for each node using PPR-Nibble. The time complexity of PPR-Nibble is $O(\frac{1}{\alpha \epsilon})$ as stated in Theorem 2, where $\alpha$ is the return probability for PPR and $\epsilon$ is the approximation error bound for Approximate-PPR. In this work, by setting $\alpha=0.15$ and setting $\epsilon$ to 0.1 or 0.01, the local clustering algorithm almost runs in constant time, which is quite fast. In the meanwhile, we can use multiprocessing to speed up preprocessing for all nodes in the graph. Actually, the largest dataset (ogbn-papers100M) can be preprocessed in several hours.
>
> **[Regarding Dynamic Graphs]** Thanks for this great question. It would be interesting to explore local clustering in the streaming or dynamic setting. But it is beyond the scope of this paper. It could be a very interesting future work.
>
> **[Regarding the Number of GPUs for experiments]** We conduct our experiments on a server with 8 NVIDIA Tesla V100. For the ogbn-papers100M (the 100M nodes and 1B edges we mentioned in the abstract), we only use a **single** GPU for training. For the link prediction datasets (such as ogbl-ppa, ogbl-citation), as the samples are node pairs and the number of such pairs is very large, we use multi-GPU to speed up training although single GPU training is still affordable.
>
> **[Regarding Typos]** Thank you for pointing out these typos. We will update them in the next revision.

---

### Official Review · AnonReviewer4 · 2020-10-31
**Official Blind Review #4**

**Rating:** 6
**Confidence:** 3

**Review:**

##########################################################################

Summary:
This paper proposes to utilize local clustering to efficiently search for small but compact subgraphs for Graph Neural Networks (GNN) training and inference. The existing PPR-Nibble is adopted for the local clustering search. Experiments validate the effectiveness of the proposed method.

##########################################################################

Pros:
+ The paper is clear and well organized.
+ The idea of using local clustering to design a lightweight, effective and scalable GNN framework is interesting.
+ The analysis of the connection between GNNs and local clustering is also interesting.
+ Extensive experiments are conducted to validate the effectiveness of the proposed method.

##########################################################################

Cons:
- One major concern about the paper is the lack of novelty. Although the idea of using local clustering is very interesting, it is straightforward to apply an existing local clustering algorithm into GNN especially considering the existing methods that utilize the global graph partition.

- In the introduction, it is claimed that the graph sampling techniques may lead to the high variance issue, while from the experiment, we can see that compared to SOTA sampling methods (e.g. GraphSAINT), the proposed method has larger variance actually. The reason for this should be given.

- The advantage of the proposed method over the methods based on global graph partition (e.g., ClusterGCN) is not well illustrated.

- In Table 6, only the result with "LCGNN-Transformer" is provided, and it is not clear whether the improvement is caused by the transformer encoder. The comparison of the learning paradigm is missing. For better comparison, especially with ClusterGCN, the result of "LCGNN-GCN" is expected.

Minor issues:
- In the paper, it is claimed that "Compared to full-batch GNNs, sampling-based GNNs and graph partition-based GNNs, LCGNN performs comparably or even better...", while from the experiments, we can see that it can be worse for some tasks. Moreover, the improvement may be not mainly caused by LCGNN itself. It is better to make this clearer.

##########################################################################

Questions during the rebuttal period:
Please address and clarify the cons above.

---

> ### Author Response · Authors · 2020-11-24
> **Response to AnonReviewer4**
>
> We appreciate your positive feedback and will revise our submission accordingly.
>
> **[Regarding Novelty]** Our contributions are two folds. First, we establish a theoretical connection between GNNs and local clustering. Second, we propose a local clustering GNN framework inspired by the theoretical connections, which is able to scale up to very large graphs effectively and efficiently. We believe that a simple but effective method is what we need in practice when applying GNN to real-world large-scale problems.
>
> **[Regarding the High Variance Issue of Graph Sampling]** When we introduced the high variance problem, our focus was the Neighborhood Sampling method. Our original texts are:
> > “For instance, graph sampling techniques, such as neighborhood sampling in GraphSAGE (Hamilton et al., 2017), may lead to the high variance issue (Chen et al., 2018a).”
>
> > “As a randomized algorithm, Neighborhood Sampling was further improved by FastGCN (Chen et al., 2018b), Stochastic GCN (Chen et al., 2018a) and Adaptive Sampling (Huang et al., 2018) for variance reduction.”
>
> Moreover, we want to point out that the variance in Section 6 cannot be compared directly. The main reason is that different sampling-based methods use different sample sizes. It would be interesting to further study the variance of different methods, although we focus on designing scalable GNN solutions and the variance issue is not the main purpose of this work.
>
> **[Regarding the Comparison to ClusterGCN]** From the OGB leaderboards, our proposed LCGNN-Transformer performs better than ClusterGCN. For example, LCGNN-Transformer achieves 0.8131 in ogbn-products, significantly better than ClusterGCN (SAGE aggr)’s 0.7897. Although one may think ClusterGCN can also replace its encoder with a Transformer for better performance, we want to emphasize that it would be much more expensive. Take ogbn-products as an example, the ClusterGCN reported in the leaderboard (https://github.com/snap-stanford/ogb/blob/master/examples/nodeproppred/products/cluster_gcn.py) partitions the input graph into 15,000 subgraphs. Each subgraph, on average, contains 163 nodes, which is 2.5 times of the cluster size used in LCGNN-Transformer. Due to the quadratic memory of the Transformer, the memory consumption of ClusterGCN (if use Transformer encoder) could be 2.5 * 2.5 = 6.5x of the memory used by LCGNN-Transformer. The memory consumption could be crazier if we scale to larger graphs such as ogbn-papers100M.
>
> **[Regarding Table 6]** Thanks for the great suggestion. We will add the result of “LCGNN-GCN” in the recent revision. The main reason is that the training of ogbl-citation is quite time-consuming.

---

### Official Review · AnonReviewer2 · 2020-11-01
**interesting work, but needs more study**

**Rating:** 5
**Confidence:** 4

**Review:**

This is an interesting work on GCN. The idea is to form local graph for each node using PPR-Nibble, a local clustering method proposed before, and then use transformer on top of the local graph as encoder for node classification and link prediction. The algorithm is simple and easy to understand. I have several concerns for this paper:

1) It seems that the local clustering+transformer encoder performs best, while it is hard to tell whether local clustering is the key to this improvement. In Table 2, local clustering + GCN encoder performs even worse than most of GCN methods, which might indicate the local clustering is not useful. In other words, what if other comparing GCN methods are also using transformer encoder? Without this study, it is hard to tell what is the novelty of this paper and whether the local clustering is useful.

2) What is the time complexity of this method? It seems the proposed method is very computation expensive. As it needs to form transformer for each node in the graph, not even to mention the time for PPR-Nibble for each node as well. This method might be even slower than GCN itself.

3) All the theorems are not derived by this paper, but from other papers on PPR-Nibble. Will these theorems have assumptions about the underlying graphs? And will this theorem works for the graphs in the experiment parts? Do the proposed method only work with nodes with small numbers of neighbors?

4)  Besides PPR-Nibble, there are many local clustering methods or even using on-line clustering so that the whole graph is not needed to fit into the memory to form the clustering, it will be interesting to see how various clustering methods work under the framework.

---

> ### Author Response · Authors · 2020-11-24
> **Response to AnonReviewer2**
>
> Thank you for your comments. We would also like to clarify a couple of points.
>
> **[Regarding Q1 about Table 2 and the usefulness of Local Clustering]** In Table 2, the performance of LCGNN-GCN is better than the full-batch GCN (0.7683 v.s. 0.7564). LCGNN-SAGE performs competitively to full-batch GraphSAGE (0.7858 v.s. 0.7850) and GraphSAINT (SAGE aggr) (0.7858 v.s. 0.7908). LCGNN-GAT performs better than NeighborSampling (GAT aggr) (0.8080 v.s. 0.7945) and GraphSAINT (GAT aggr) (0.8080 v.s. 0.8027). The above comparison illustrates the usefulness of Local Clustering, with the same encoder, local clustering based methods are comparable or even better than their full-batch/NeighborSampling/GraphSAINT counterparts.
>
> **[Regarding Q1 about Other Methods Using Transformer as the Encoder]** The reviewer points out the possibility that other comparing GCN methods are also using the transformer encoder. We want to emphasize it would be very time/memory expensive due to the notoriously high cost of Transformers. For example, suppose one wants to run a transformer on ogbn-products graph, then the Transformer encoder has to compute a 2,449,029 by 2,449,029 attention matrix, which is unaffordable. The advantage of LCGNN is the lightweightness of local clustering, which enables efficient small subgraphs extraction so that we can use powerful (but expensive) transformer encoders for graph representation learning.
>
> **[Regarding Complexity]** We preprocess the local clusters for each node using PPR-Nibble. The time complexity of PPR-Nibble is $O(\frac{1}{\alpha \epsilon})$ as stated in Theorem 2, where $\alpha$ is the return probability for PPR and $\epsilon$ is the approximation error bound for Approximate-PPR. In this work, by setting $\alpha=0.15$ and setting $\epsilon$ to 0.1 or 0.01, the local clustering algorithm almost runs in constant time, which is quite fast. In the meanwhile, we can use multiprocessing to speed up preprocessing for all nodes in the graph. Actually, the largest dataset (ogbn-papers100M) can be preprocessed in several hours.
>
> **[Regarding Assumptions of Theorems and Their Application in LCGNN]** There are no additional assumptions about the theorems used in this paper. When applying those theorems in LCGNN, the only assumption is the existence of a low-conductance cluster for every node (or most nodes) in the graph. We discuss this assumption in the last paragraph of Section 4, which is believed to be true for most real-world social and information networks. The worst case, for example, can be a fully-connected graph, where there do not exist local clusters.
>
> **[Regarding Online Local Clustering]** Thanks for this great question. It would be interesting to explore local clustering in the streaming or dynamic setting. But it is beyond the scope of this paper. It could be a very interesting future work.

---

### Decision · Program_Chairs · 2021-01-07
**Final Decision**

**Decision:**

Reject

**Comment:**

The paper considers using local spectral graph clustering methods such at the PPR-Nibble method for graph neural networks.  These local spectral methods are widely used in social networks, and understanding neural networks from them is interesting.

In many ways, the results are interesting and novel, and they deserve to be more widely known, but there are several directions to make the work more useful to the community.   These are outlined in the reviewer comments, which the authors answered partially but not completely satisfactorily.  Much of this has to do with explaining how/where these (these very fundamental and ubiquitous) methods are useful in a particular application (GNNs here, and node embeddings below).  An example of a paper that successfully did this is "LASAGNE: Locality And Structure Aware Graph Node Embedding, E. Faerman, et al.  Proc. 2018 Conference on Web Intelligence."  (That is mentioned not since it is directly relevant to this paper, but since it provides an example of how to present the use of a method such as PPR-Nibble for the community.